# Reforestation Increases the Aggregate Organic Carbon Concentration Induced by Soil Microorganisms in a Degraded Red Soil, Subtropical China

**DOI:** 10.3390/microorganisms11082008

**Published:** 2023-08-04

**Authors:** Yunfang Ke, Hui Li, Tianyu Luo, Baodong Chen, Qiong Wang, Xueru Jiang, Wei Liu

**Affiliations:** 1Jiangxi Provincial Key Laboratory of Silviculture, Jiangxi Agricultural University, Nanchang 330045, China; keyunfang9867@163.com (Y.K.);; 2College of Forestry/College of Art and Landscape, Jiangxi Agricultural University, Nanchang, 330045, China; 3Research Center for Eco-Environmental Sciences, Chinese Academy of Sciences, Beijing 100085, China; 4University of Chinese Academy of Sciences, Beijing 100049, China; 5Key Laboratory of Poyang Lake Watershed Agricultural Resources and Ecology of Jiangxi Province, Jiangxi Agricultural University, Nanchang 330045, China

**Keywords:** degraded red soil, reforestation, aggregate organic carbon, soil microorganisms

## Abstract

In the process of biological carbon (C) sequestration during reforestation in degraded red soil, due to the decomposition of soil microorganisms, the interaction between soil organic carbon (SOC) and aggregates has an important effect on soil C sequestration. In this study, six common reforestation models and three soil layers were selected in a degraded red soil area of the central subtropical region to determine the composition of soil aggregates and the distribution of SOC in soil aggregates. Based on the results of the soil physicochemical properties and microbial community composition biomass, we assessed the changes in aggregate-associated organic C storage during fluctuations in the stability of the aggregates. After reforestation, the SOC stock increased by 131.28–140.00%. Compared with the three pure forests and broad-leaved mixed forests, coniferous and broad-leaved mixed forests showed the largest proportion of macroaggregates (85.48–89.37%) and higher SOC accumulation. Soil microbial biomass mainly affected the decomposition process of SOC by affecting the stability of the soil aggregates, and the effect of bacteria was more significant. Coniferous and broad-leaved mixed forests can provide more soil microorganisms and C sources than pure forest, thus promoting macroaggregate formation and stability and related organic C storage. This reforestation model has greater C sequestration potential.

## 1. Introduction

At present, the world emits approximately 50 billion metric tons of greenhouse gases into the atmosphere every year, and global climate change poses a major threat to human society [1]. To avoid climate disasters, 195 countries agreed to the ‘Paris Agreement’ [2], which established the overall goal of a global response to climate change threats: global carbon emissions need to be reduced by half by 2030 and must reach ‘net zero’ (carbon neutrality) by the middle of the 21st century [2,3]. The ‘carbon neutrality’ goal indicates China’s efforts to achieve ‘zero emissions’ of carbon dioxide (CO_2_) by 2060 [4]. It is necessary to improve carbon (C) sequestration to remove CO_2_ through ecological construction, soil C sequestration, utilization, and storage projects and technologies [5,6]. Vincevica-Gaile proposed some methods of stable and circular economy by studying peatland degradation [7]. In the past decade, people have paid increasing attention to natural solutions, and biological C sequestration is the most promising method with which to alleviate global warming [8]. Soil C sequestration is the most effective natural strategy able to mitigate the increase in the atmospheric CO_2_ concentration due to fossil fuel combustion and vegetation change [9].

The performance of carbon sequestration by terrestrial ecosystems is the most economically feasible and environmentally friendly way to slow the increase in the atmospheric CO_2_ concentration [10,11]. As the main component of terrestrial ecosystems, forest soil C storage in China ranges from 44 t·hm^−2^ to 264 t·hm^−2^, with an average of 107.80 t·hm^−2^ [12]. Forests also have the largest ‘carbon pool’ on land and play an important role in regulating the climate and alleviating global warming [13]. However, the threshold of forests, as the largest C sink among terrestrial ecosystems, has not been scientifically determined [14]. Red soil, whose forest area accounts for 45% of the national forest area, is an important soil resource in China [15]. Due to regional ecological vulnerability and human disturbance, the subtropical red soil area has become an area with some of the most serious erosion and soil degradation in China [16]. The loss of SOC caused by soil erosion is serious, with a large erosion area and high intensity [17]. SOC loss mainly occurs in the surface layer of soil [18], and its dynamic change affects soil fertility and the CO_2_ content in the atmosphere, thus having an important impact on surface vegetation and climate change [19]. Reforestation can effectively reduce runoff and soil erosion, improve soil C sequestration in red soil forest ecosystems, and reduce the increase in the atmospheric CO_2_ concentration economically and efficiently, making reforestation the key method for restoring ecosystems [20]. In response to the ‘double carbon’ policy, we should make full use of the forest soil C pool and adopt the most promising biological C sequestration method in reforestation to slow the rate of red soil erosion and alleviate global warming.

Although most of the C stored in soil can be sequestered for a long time [21,22], forest management methods, such as reforestation models, will cause significant indigenous differences in the SOC [23,24]. For example, higher tree species diversity in forests contributes to more SOC storage [25,26]. Different input [27] and decomposition rates [6,28] are considered to be the most likely factors affecting the effects of tree species on soil C storage. Therefore, to make targeted use of tree species to link SOC, we urgently need to determine which processes control C pool differences and to study the forms and stability of SOC. The decomposition or preservation of SOC is related to the contact of decomposers and extracellular enzymes to SOC [29]. The role of soil microorganisms in the process of SOC operation in the forest soil ecosystem is very important. It can be said that soil microorganisms are the core link of the carbon conversion process in the forest ecosystem. Its composition characteristics (such as community composition, diversity, etc.) can affect the soil biofeedback regulation of the forest C cycle [30].

The forms and stability of C are related to the decomposition or preservation of SOC [31], which is determined by the spatial distribution (aggregate distribution) of SOC in the soil matrix [32]. Soil aggregates, as the basic unit of soil structure, are the most important mechanism for protecting SOC from microbial decomposition and maintaining its stability, even exceeding the influence of SOC decomposition itself [33]. The stability of soil aggregates and the distribution of aggregate organic C are closely related to the size and stability of the SOC pool, which can explain the variation characteristics and possible mechanisms of the SOC pool [34]. Although studies have focused on the changes in SOC and its components, aggregate stability, and aggregate-bound SOC under different tree species in degraded red soil, there is still a lack of research on SOC storage mechanism from the perspective of aggregate organic carbon distribution and microbial drive.

In this study, six common reforestation models were applied in a degraded red soil region of the central subtropical region. These models were used to study the effects of reforestation on soil microbial biomass, the composition of soil aggregates and the distribution of SOC in soil aggregates. We tested the following hypotheses:

**Hypothesis 1 (H1).** *As vegetation growth is conducive to soil and water conservation, macroaggregates increase, and the stability of the soil structure improves during reforestation in degraded red soil regions*.

**Hypothesis 2 (H2).** *As surface soil is more affected by external factors, the contents of macroaggregates and aggregate organic C are significantly higher in surface soil than in deep soil*.

**Hypothesis 3 (H3).** *As the coniferous and broad-leaved mixed forest has richer litter and root composition, there is a greater proportion of macroaggregate organic C stabilized by microbial activities compared to other models*.

We then explained how afforestation types affect soil microbial-induced aggregate organic carbon distribution and SOC fixation. The goal is to provide reference materials for soil quality evaluation and the soil erosion resistance of reforestation in degraded red soil regions, providing theoretical support for biological C fixation in forest ecosystems.

## 2. Materials and Methods

### 2.1. Research Sites and Sampling Points

The study area is located in Taihe County, Jiangxi Province, South China (114°49′44″ E, 26°54′47″ N), which is characterized by a subtropical humid monsoon climate. The average annual precipitation is 1726 mm, concentrated from April to June. The average annual temperature is 18.60 °C, and the soil type is red or iron (FAO/UNESCO) [35]. *Pinus massoniana*, *Liquidambar formosana*, *Schima superba* and *Cunninghamia lanceolata* were planted in the vegetation restoration experiment. Since 1990, Jiangxi Agricultural University has carried out reforestation experiments using different degrees of degradation and types of red soil in 9 counties (cities), and has established a test base with a total area of 640 hm^2^. In 30 years of continuous research, the relevant teams have explored the reforestation model and mechanism in degraded red soil areas from the aspects of soil structure, physicochemical properties, moisture, microbial and enzyme activities, the fine roots of forest ecosystems, and plant diversity. However, there is still no research on the relationship and the effect of the interaction among soil microorganisms, soil structure improvement and the C sequestration capacity on the restoration of ecosystem functions in degraded red soil areas.

Under the guidance of field research in June 2021, five reforestation models were selected as research objects, and bare land without vegetation restoration was chosen as the control (Figure 1). The models included pure *Schima superba* (SS) forest, pure *Liquidambar formosana* (LF) forest, pure *Pinus massoniana* (PM) forest, broad-leaved mixed forest of *Liquidambar formosana* and *Schima superba* (LF × SS), coniferous and broad-leaved mixed forest of *Pinus massoniana*, *Liquidambar formosana* and *Schima superba* (LF × SS × PM), and bare land (control, CK). Under the premise of ensuring that the typical spacing between the pure forest and mixed forest was greater than 1 km and that the repeated spacing of the same tree species was greater than 100 m, we selected 4 plots (20 m × 20 m) for each reforestation type. In our study, there was a total of 24 plots. Through field investigation, the basic status of the statistical research plots is shown in Table 1.

### 2.2. Soil Sampling Method

Before the profile was dug, the vegetation and litter on the soil surface were removed, and the sampling depth was divided into 0–10 cm (surface layer), 10–20 cm (middle layer) and 20–40 cm (deep layer), with 3 replicates each. During the collection process, the original soil structure was maintained and loaded into stainless steel boxes. For each sample, 1.00 kg was collected, for a total of 216 samples. The soil chemical and physical properties of the study sites are provided in Table 2.

The collected soil samples were divided into two parts. The first set of samples was used to measure the soil physical and chemical properties, and to analyze the soil aggregates after natural air drying, and the second set of samples was stored at −20 °C for soil microbial biomass determination and community structure analysis. The method used for soil aggregate analysis was as follows [36]: the undisturbed soil was gently stripped into small soil blocks with a diameter of >8 mm along the natural structure in the laboratory, and spread flat in a ventilated place and naturally dried.

### 2.3. Determination of Soil Physicochemical Properties

The soil moisture (%) was determined using the oven-drying method (105 °C, 48 h drying to constant weight) [37]. The soil bulk density (BD) was measured using the ring knife method. The soil pH was measured using a composite electrode pH meter (PB-10; Sartorius AG, Gottingen, Germany) (soil/distilled water *v*:*v* = 1:2.5). The soil total nitrogen (TN) was determined using a continuous flow automatic analyzer (automatic analyzer III, Bran + Luebbe GmbH, Berlin, Germany).

The soil mechanical stability (MS) and water-stable aggregates (WS) [38] were determined using the dry sieving method and wet sieving method, respectively [39]. Two hundred grams of air-dried soil was sieved through a set of sieves with apertures of 2, 1, 0.5, 0.25 and 0.053 mm in turn, and the percentages of aggregates at all levels of the dry sieve were weighed and calculated. Then, each 100 g soil sample was prepared according to the proportion of each particle size and laid flat on the set of sieves for wet screening. After 10 min of infiltration with deionized water, the soil samples were manually sieved 20 times, with an amplitude of 3 cm. After vibration, the sieve was removed, and the aggregates left on the sieve were washed into an aluminum box with water. After settlement, the supernatant was discarded, and the sample was dried at 60 °C and weighed to obtain the percentages of water-stable large macroaggregates (>2 mm aggregates), middle macroaggregates (1–2 mm aggregates), small macroaggregates (0.25–1 mm aggregates), microaggregates (0.053–0.25 mm aggregates) and clayey silt aggregates (<0.053 mm aggregates).

The SOC and soil aggregate organic C were determined using the dichromate oxidation method: about 0.20 g of air-dried soil was weighed and sieved through a 0.149 mm sieve; then, 5 mL of 0.80 mol·L^−1^ potassium dichromate and 5 mL of H_2_SO_4_ were added, and placed in an oil bath preheated to 185 °C for 7 min. Two drops of phenanthroline indicator were added, and the SOC content was calculated via the consumption of 0.20 mol·L^−1^ FeSO_4_ [40]. Corresponding to the aggregates of each particle size, the organic carbon sizes of the soil aggregates of each particle size were large macroaggregate organic carbon (>2 mm aggregate-C), middle macroaggregate organic carbon (1–2 mm aggregate-C), small macroaggregate organic carbon (0.25–1 mm aggregate-C), microaggregate organic carbon (0.053–0.25 mm aggregate-C) and clayey silt aggregate organic carbon (<0.053 mm aggregate-C).

### 2.4. Soil Microbial Biomass and Community Structure

In this study, phospholipid fatty acid (PLFA) spectrum analysis was used to characterize the soil microbial biomass and community structure, and esterified c19:0 was used as the internal standard for determination via gas chromatography–mass spectrometry. The abundance of individual fatty acids was determined using nmol, relative to each gram of dry soil and using standard nomenclature. Referring to previous research methods, this study used fatty acids such as 15:0, i15:0, a15:0, i16:0, 17:0, a17:0, i17:0, 16:1ω7, cy17:0, 18:1ω7 and cy19:0 to characterize the bacterial biomass. The fungal biomass was characterized using 18:2ω6,9 fatty acids. The arbuscular mycorrhizal fungi (AMF) biomass was characterized using 16:1ω5 fatty acids, and protozoa biomass was characterized using 20:4ω6 fatty acids.

### 2.5. Method of Calculation

#### 2.5.1. Calculation of Stability Parameters of Soil Water-Stable Aggregates

In this study, the stability of soil water-stable aggregates was analyzed using the following four indicators: the mass percentage *w_i_* of water-stable aggregates with different particle sizes was calculated using Formula (1); the content of >0.25 mm stable aggregates (DR_0.25_) was calculated using Formula (2); and the mean weight diameter (MWD) and geometric mean diameter (GMD) were calculated using Formulas (3)–(4), respectively [41]. The fractal dimension (D) [42] was calculated using Equation (5), and the logarithm of both sides was used to obtain Equation (6). The equations are as follows:(1)$$w_{i}=\frac{M_{wi}}{\left(M_{w1}+M_{w2}+M_{w3}+M_{w4}+M_{w5}\right)}\times 100\%\ldots$$
(2)$${\mathrm{DR}}_{0.25}=\frac{M_{r>0.25}}{M_{T}}\times 100\%\ldots$$
(3)$$\mathrm{MWD}=\frac{\sum_{i=1}^{n}(\bar{R_{i}}\times w_{i})}{\sum_{i=1}^{n}w_{i}}\ldots$$
(4)$$\mathrm{GMD}=\exp\left(\frac{\sum_{i=1}^{n}\left(w_{i}\times \ln\bar{R_{i}}\right)}{\sum_{i=1}^{n}w_{i}}\right)\ldots$$
(5)$$\frac{M(r<\bar{R_{i}})}{M_{T}}={\left(\frac{\bar{R_{i}}}{R_{max}}\right)}^{3-D}\ldots$$
(6)$$\mathrm{lg}\left[\frac{M\left(r<\bar{R_{i}}\right)}{M_{T}}\right]=\left(3-D\right)\mathrm{lg}\left(\frac{\bar{R_{i}}}{R_{max}}\right)\ldots$$

In the above formulas, $w_{i}$ is the mass percentage of the *i* class aggregates (%); *M_wi_* is the mass of the aggregates at each particle level (g); $\bar{R_{i}}$ is the average diameter of the class of aggregates screened (mm); *M_T_* is the total mass of each aggregate (g); *R_max_* is the maximum particle size of the aggregates (mm); and *M*(*r <*
$\bar{R_{i}}$) is the aggregate mass with a particle size less than $\bar{R_{i}}$ (g).

#### 2.5.2. SOC Storage and Contribution Rate of Aggregate Organic Carbon

The SOC stock (SOC-stock, kg·m^−2^) formula [43] is as follows:(7)$$\mathrm{SOC-stock}=\frac{SD\times BD\times \mathrm{SOC}}{100}\ldots$$

The calculation formula for the SOC stock (stock of OC*i*, g·m^−2^) of the soil aggregate fractions [44] is as follows:(8)$${\mathrm{Stock}\;\mathrm{of}\;\mathrm{OC}}_{i}=\frac{SD\times BD\times w_{i}\times {\mathrm{OC}}_{i}}{10}\ldots$$

In the above formulas, *SD* represents the soil depth (cm); *BD* represents the soil bulk density (g·cm^−3^); SOC represents the SOC content (g·kg^−1^); $w_{i}$ represents the percentage of the aggregate mass at level *i* (%); and ${\mathrm{OC}}_{i}$ represents the SOC content of class *i* aggregates (g·kg^−1^).

### 2.6. Data Analysis

Data analysis was performed in SPSS 22. One-way ANOVA was used to test the differences in the weight percentage, MWD, GMD, D, content of stable aggregates > 0.25 mm (DR_0.25_), SOC content and storage of each aggregate component under the different reforestation modes during the reforestation of degraded red soil. All abbreviations and full names appearing in this study were shown in Table 3, which correspond to the units used. A correlation test was used to test the interrelationship between the aggregate structure characteristics and the soil physicochemical properties, and between the distribution of organic carbon in the aggregates and the soil microbial biomass in different soil layers. Before the correlation analysis, all the data were tested using a box plot to check whether there was a difference. For the constant, if there was an exception, the correlation analysis was performed by replacing it with the mean, and the number was judged by drawing a scatter plot. If the data conformed to the linear correlation, Pearson correlation analysis was used. If not, Spearman rank correlation analysis was considered according to the situation. After the analysis method was determined, the normal distribution test was performed on all data, and *p* > 0.05 was consistent with the normal distribution to continue the analysis. The above analysis was performed in SPSS 22 (IBM^®^SPSS^®^ Statistics, IBM, New York, NY, USA). The relevant figures were produced using Origin 2018 and R software (version 4.2.0).

## 3. Results

### 3.1. Effects of Different Reforestation Models on the Mass Distribution and Stability of Soil Aggregates

Overall, the mass% of each particle size aggregate in the five reforestation models showed an ‘N’ trend with decreasing particle size (Figure 2). Except for the CK, the proportion of soil macroaggregates (>0.25 mm) in each soil layer of the other models was greater than 80%, and the proportion of <0.053 mm aggregates was the smallest, with values of 3.20–14.61%. At the 0–10 cm depth, PM had the greatest mass% (42.69%) of >2 mm aggregates, while LF × SS × PM had the greatest mass% (31.08%) of 1–2 mm aggregates (Figure 2a). However, at depths of 10–20 cm and 20–40 cm, the mass% of the >2 mm aggregates and 1–2 mm aggregates reached the highest values in LF × SS × PM and SS (41.84%, 37.50%, and 24.45%, 21.94%, respectively) (Figure 2b,c).

With the increase in soil depth, except for LF × SS × PM, the mass% of the >2 mm aggregates and 1–2 mm aggregates of the reforestation models increased significantly, while the proportions of the 0.053–0.25 mm aggregates and <0.053 mm aggregates decreased significantly. However, the >2 mm aggregates of LF × SS × PM in the 10–20 cm soil layer were significantly higher than those in the surface soil (*p* < 0.01).

The DR_0.25_, MWD and GMD of the five reforestation models were significantly higher than those of the CK (Figure 3). At the 0–10 cm depth, the MWD, GMD and DR_0.25_ values of LF and LF × SS × PM were higher, while at 10–20 cm and 20–40 cm, they were the greatest for LF × SS × PM (Figure 3). Overall, except for CK, SS had the lowest MWD and GMD. The MWD and GMD of each soil layer in LF × SS × PM first increased and then decreased with increasing soil depth (Figure 3a,b). The D of the five reforestation models decreased gradually with soil depth, with the lowest level in LF × SS × PM and the highest in CK (Figure 3d). This result shows that the soil textures of LF, SS, PM and LF × SS tended to be consistent, and the 20–40 cm depth soil texture of LF × SS × PM was greater than those of 0–10 cm and 10–20 cm (Figure 3). Based on the degree of soil texture uniformity, the CK soil structure was poor, while the LF × SS × PM soil structure was the best.

### 3.2. Correlation between Soil Aggregates and Physicochemical Properties

To analyze the key factors influencing the composition and stability of the soil aggregate in the process of reforestation in degraded red soil, the correlations between the soil moisture, pH, bulk density, TN, SOC and aggregate composition and stability parameters were analyzed (Figure 4). SOC storage was significantly positively correlated with >2 mm aggregates (*p* < 0.01), but significantly negatively correlated with 0.053–0.25 mm aggregates (*p* < 0.01) and <0.053 mm aggregates (*p* < 0.01). TN, C:N and SOC were significantly positively correlated with >2 mm aggregates at the 0–10 cm (*p* < 0.05) and 20–40 cm (*p* < 0.01) depths, but negatively correlated with 1–2 mm aggregates at the 10–20 cm depth (*p* < 0.01).

In terms of the soil aggregate composition and stability parameters, the MWD, GMD and DR_0.25_ at the 0–10 cm depth were significantly positively correlated with the mass of the >2 mm aggregate and 1–2 mm aggregate (*p* < 0.01), but significantly negatively correlated with the 0.25–1 mm aggregate, 0.053–0.25 mm aggregate and <0.053 mm aggregate (*p* < 0.01), with an inverted result for D (*p* < 0.01).

### 3.3. Effects of Different Reforestation Models on the Distribution of the Soil Aggregate Organic C Concentration and Storage

After 30 years of reforestation, the SOC concentration (Table 1) and the soil aggregate organic C concentration increased (Figure 5). The soil aggregate organic C concentration of the five reforestation models presented an ‘M’-type change and reached the lowest level in the microaggregates. The soil aggregate organic C concentration of SS at the 0–10 cm depth was higher than that of the other models (*p* < 0.01), and the >2 mm aggregate-C concentration of LF × SS × PM was the lowest (Figure 5a). However, at depths of 10–20 cm and 20–40 cm, the highest concentrations of the 1–2 mm aggregate-C, 0.25–1 mm aggregate-C, 0.053–0.25 mm aggregate-C, and <0.053 mm aggregate-C were found in LF × SS × PM (Figure 5b,c). With increasing soil depth, the soil aggregate organic C concentration in the five reforestation models decreased, but it was still higher than that of the CK. Moreover, the concentrations of the >2 mm aggregate-C and < 0.053 mm aggregate-C were lower than those of the other particle size aggregates.

After 30 years of forest restoration in degraded red soil, the soil aggregate organic C storage increased obviously due to the lowest aggregate organic C stocks in the CK (Figure 6). At the 0–10 cm and 10–20 cm depths, the soil aggregate organic C storage of SS was higher, and the aggregate organic C storage of LF and PM was lower (Figure 6a,b). However, LF × SS × PM had the highest 0.25–1 mm aggregate-C storage at depths of 10–20 cm and 20–40 cm (Figure 6b,c). Based on the distribution of SOC storage in the aggregates, SOC was mainly stored in macroaggregates, with a proportion of 85.15–89.02%. The <0.053 mm aggregate-C storage was the lowest, with a proportion of only 14.40–17.22% (Figure 6). The soil aggregate organic C storage of LF, PM, and LF × SS in each soil layer was equivalent, and was only 65% of that of SS.

### 3.4. Effects of Different Reforestation Models on the Soil Microbial Community Biomass

The microbial community biomass of the six reforestation models decreased gradually as the soil depth increased, and that of the 0–10 cm soil layer was significantly greater than that of the other soil layers (*p* < 0.05) (Figure 7). In the 0–10 cm soil layer, the total bacterial biomass of SS and LF × SS × PM was significantly higher than that of CK, and there was no significant difference between LF, PM, LF × SS and CK (*p* < 0.05) (Figure 7a). The total fungal biomass of LF × SS × PM was significantly higher than that of the other models (*p* < 0.05) (Figure 7b). The AMF biomass of SS was the highest (0.29 nmol·g^−1^), and that of LF was the lowest (0.18 nmol·g^−1^) (Figure 7c). The protozoan biomass of LF×SS was the highest (0.13 nmol·g^−1^), and that of LF × SS × PM was the lowest (0.03 nmol·g^−1^) (Figure 7d). In the 20–40 cm soil layer, the total fungal and AMF biomasses of SS were significantly higher than those of the other models (*p* < 0.05). The protozoan biomass of LF × SS × PM was significantly higher than that of the other models (*p* < 0.01).

To analyze the key effects of soil microorganisms on aggregate organic C during the reforestation of degraded red soil, the correlation between soil bacteria, fungi, AMF and protozoa and the distribution of SOC in aggregates were analyzed (Figure 8). Bacteria and AMF were significantly correlated with the distribution of organic carbon in ≤2 mm aggregates, while fungi was significantly correlated with the distribution of organic carbon in >1 mm aggregates in the 0–10 cm soil layer (*p* < 0.05). Protozoa was significantly correlated with the distribution of organic carbon in 1–2 mm, 0.25–1 mm and <0.053 mm aggregates in the 20–40 cm soil layer (*p* > 0.05).

## 4. Discussion

It is generally believed that soil water-stable aggregates are the basis for maintaining soil structural stability and can reflect the quality of the soil structure to a certain extent [45]. Previous studies have shown that a higher proportion of water-stable macroaggregates (>0.25 mm) produces a more stable soil structure [46]. In this study, >0.25 mm water-stable aggregates were dominant in the different reforestation models, and the mass ratio was greater than 80% (Figure 2), which is consistent with the research results regarding the effects of vegetation restoration on the SOC of water-stable aggregates in degraded red soil [47]. Reforestation in the degraded red soil regions of southern China contributes to the transformation of soil microaggregates into macroaggregates and improves the stability of the soil structure, which is consistent with Hypothesis H1.

The higher mass% of macroaggregates in each soil layer of the coniferous and broad-leaved mixed forest suggests (Figure 2) that improvements in the soil structure in coniferous and broad-leaved mixed forest are better after reforestation [48]. The C:N and SOC were positively correlated with the >2 mm aggregates (Figure 4), possibly because there are more shrubs and small tree species [49] beneath the coniferous and broad-leaved mixed forest. The increase in the soil nutrient content and root interpenetration improved the aggregate characteristics and soil structure stability [50]. Due to the significant negative correlation between macroaggregates and microaggregates (Figure 4), our study suggested that soil microaggregates were the transition between macroaggregates and clayey silt aggregates [51,52], which could be cemented by clayey silt aggregates and broken from macroaggregates [53,54]. The *Schima superba* forest, as a deciduous broad-leaved forest, has stronger microbial activity (Figure 7) and more SOC decomposition, which makes macroaggregates more likely to break into microaggregates [55]. Moreover, organic and inorganic colloids in microaggregates can be closely combined with fixed C [56], and are not easily decomposed and released by microorganisms; thus, the content of microaggregates in the *Schima superba* forest is the highest.

The DR_0.25_, MWD and GMD can be used to characterize the structure of soil aggregates, for which a higher value represents a higher mass% of macroaggregates, indicating that soil corrosion resistance increases as the stability of soil aggregates increases [57,58]. The higher D reflected the size of the soil particles and the uniformity of the soil texture, indicating that compacted soil is associated with more clay particles and a poor soil structure [58]. In our study, the soil structure in coniferous and broad-leaved mixed forests was better than that in the other reforestation models [52], based on the analysis of the above indicators (Figure 3). The main reason was that the high microbial activity in the coniferous and broad-leaved mixed forests (Figure 7) improved the surface soil structure through microbial decomposition [59]. The combination of a rich understory vegetation root network (Table 1) and microbial decomposition improved the soil texture. Correlation analysis showed that soil C:N and SOC storage were significantly correlated with macroaggregates (Figure 4), thus explaining the vital significance of the C content in the soil structure. The main reason for this result is that the accumulation and decomposition of surface litter increases the contents of organic matter and effective nutrients in soil [60,61].

Reforestation increased the concentration of SOC (Table 2, Figure 5). The 0.25–1 mm aggregate-C was higher (Figure 5), which was due to the aggregation of microaggregates into a large number of temporary large aggregates under the action of various binders in the soil, such as root exudates, the microbial decomposition of polysaccharides [62], and glomalin-related soil proteins produced by AMF [39]. Some scholars have used ^13^C tracer technology to reveal the transformation of organic C in soil aggregates, and the results showed that organic C was first transformed from large aggregates to microaggregates and that the turnover rate of C gradually slowed with the decrease in aggregate size [33]. In our study, there was a significant positive correlation between microorganisms and the distribution of organic carbon in <1 mm aggregates, which indicated that the larger the particle size of soil aggregates, the greater their ability to hinder the decomposition of organic carbon by microorganisms (Figure 8) [63]. Microorganisms can only decompose organic C by secreting extracellular enzymes and consuming high amounts of energy for diffusion into aggregates [38]. Reforestation is helpful to improve the protective effect of macroaggregates, reduce the decomposition of SOC by microorganisms and increase the C sink potential of forest soil [64].

After reforestation, the soil macroaggregate organic C concentration and storage increased significantly [65], which became the main form of SOC accumulation. With increasing soil depth, the soil aggregate organic C in each reforestation model decreased significantly, and its surface aggregation was obvious [66]. This result was mainly because the accumulation of litterfall, animal and plant residues in the soil surface environment created good environmental conditions [67] for the formation of organic matter in aggregates. Since the surface soil is greatly affected by the external environment and easily changes [68], the effects of reforestation on C in the surface soil were more significantly enhanced than they were in other soil layers (Figure 5 and Figure 6), which is consistent with Hypothesis H2.

In addition to different aggregates and soil depths, the reforestation models had significant differences. Compared with other models, the *Schima superba* pure forest and coniferous and broad-leaved mixed forests with a higher vegetation coverage rate maintained more macroaggregates (Table 1, Figure 2) and reduced SOC mineralization [69]. Moreover, the decomposition rate of the mixed litter of *Schima superba* and *Pinus massoniana* stimulated the activity of soil microorganisms (Figure 7), thus promoting the decomposition rate of litter [70], which increased the content of humus in the soil as well as the accumulation of SOC, which is consistent with Hypothesis H3.

In the past 30 years, the development of forest C sequestration projects in China has been dominated by C sequestration afforestation projects and supplemented by forest management [11]. The total C (including trees, woodland and understory vegetation) in China has increased from 125.06 × 10^8^ t to 214.39 × 10^8^ t. The plantation C storage growth rate is obvious, with an average annual increase of 5.05% [71]. Compared with other emission reduction methods, forest C sinks have more economic and efficient advantages and have gradually become an important alternative to CO_2_ emission reductions [72]. The irreplaceable role of forest C sinks in mitigating climate change has been widely recognized and highly valued by the international community [73]. The results of this study show that in degraded red soil, reforestation can enhance the stability of aggregates, increase the aggregate organic carbon concentration induced by soil microorganisms [74], reduce soil erosion [75], and increase forest C sequestration.

## 5. Conclusions

The effect of reforestation on soil quality improvement was remarkable, and the stability of large aggregates increased significantly. The content of soil > 0.25 mm water-stable aggregates in each reforestation model was 1.2–1.4 times that of bare land. The soil aggregate stability and SOC concentration in the 0–10 cm soil layer were significantly higher than those in the 10–20 cm and 20–40 cm layers. Comprehensive experimental results showed that compared with the *Pinus massoniana* forest, *Liquidambar formosana* forest, *Schima superba* pure forest, and *Liquidambar formosana* and *Schima superba* broad-leaved mixed forest, the reforestation model of *Liquidambar formosana*, *Schima superba* and *Pinus massoniana* coniferous and broad-leaved mixed forests was more conducive to the formation and stability of large aggregates, more able to promote soil microbial activity and more conducive to the storage of SOC in macroaggregates. Therefore, in the future reforestation of degraded red soil, the model of coniferous and broad-leaved mixed forests of *Liquidambar formosana*, *Schima superba* and *Pinus massoniana* should be prioritized.

## Figures and Tables

**Figure 1 microorganisms-11-02008-f001:**
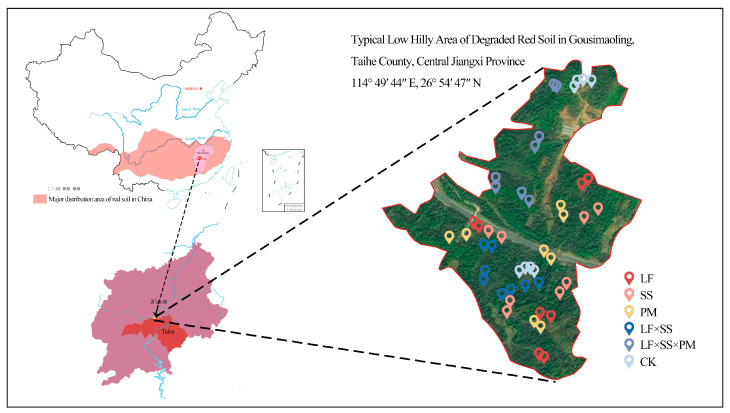
Schematic map of the sampling location and the sampling scheme. The study area is located in the degraded red soil of the low hilly area of Gousimaoling, Taihe County, Jiangxi Province, China. These stars in the figure is positioned for each sampling point. LF, *Liquidambar formosana* pure forest; SS, *Schima superba* pure forest; PM, *Pinus massoniana* pure forest; LF × SS, mixed forest of *Liquidambar formosana* and *Schima superba*; LF × SS × PM, mixed forest of *Liquidambar formosana*, *Schima superba* and *Pinus massoniana*; CK, bare land. The labels in the following figures are the same.

**Figure 2 microorganisms-11-02008-f002:**
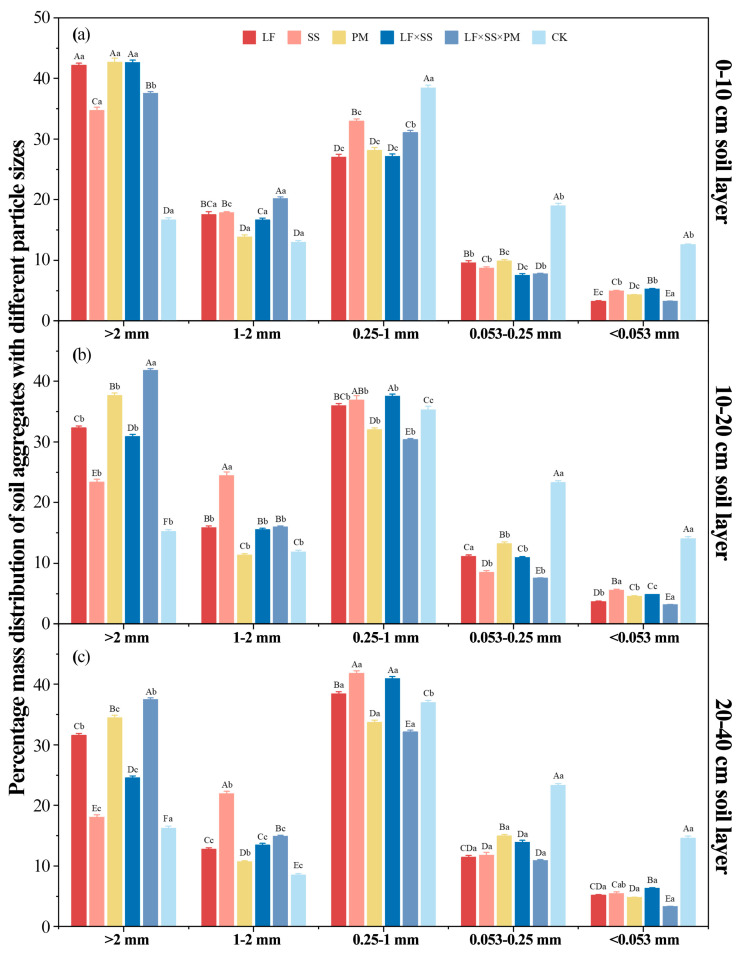
Differences in the soil aggregates at the 0–10 cm (**a**), 10–20 cm (**b**) and 20–40 cm (**c**) depths in five forest reforestation models and the CK in subtropical degraded red soil, showing the mass percentages of each particle size aggregate. Different uppercase letters indicate differences between different tree species in the same soil layer, and different lowercase letters indicate differences between different soil layers in the same forest reforestation model (*p* < 0.05). LF, *Liquidambar formosana* pure forest; SS, *Schima superba* pure forest; PM, *Pinus massoniana* pure forest; LF × SS, mixed forest of *Liquidambar formosana* and *Schima superba*; LF × SS × PM, mixed forest of *Liquidambar formosana*, *Schima superba* and *Pinus massoniana*; CK, bare land.

**Figure 3 microorganisms-11-02008-f003:**
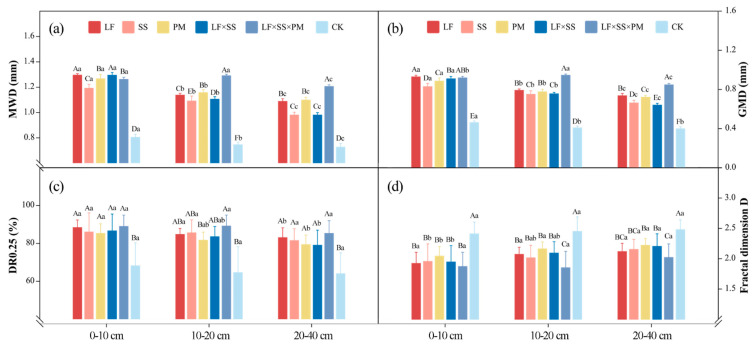
Differences in the mean weight diameter (MWD) (**a**), geometric mean diameter (GMD) (**b**), content of macroaggregates (DR_0.25_) (**c**), and fractal dimension D (**d**) at the 0–10 cm, 10–20 cm and 20–40 cm depths in five forest reforestation models and the CK in subtropical degraded red soil, showing the mass percentages of each particle size aggregate. Different uppercase letters indicate differences between different tree species in the same soil layer, and different lowercase letters indicate differences between different soil layers in the same forest reforestation model (*p* < 0.05). LF, *Liquidambar formosana* pure forest; SS, *Schima superba* pure forest; PM, *Pinus massoniana* pure forest; LF × SS, mixed forest of *Liquidambar formosana* and *Schima superba*; LF × SS × PM, mixed forest of *Liquidambar formosana*, *Schima superba* and *Pinus massoniana*; CK, bare land.

**Figure 4 microorganisms-11-02008-f004:**
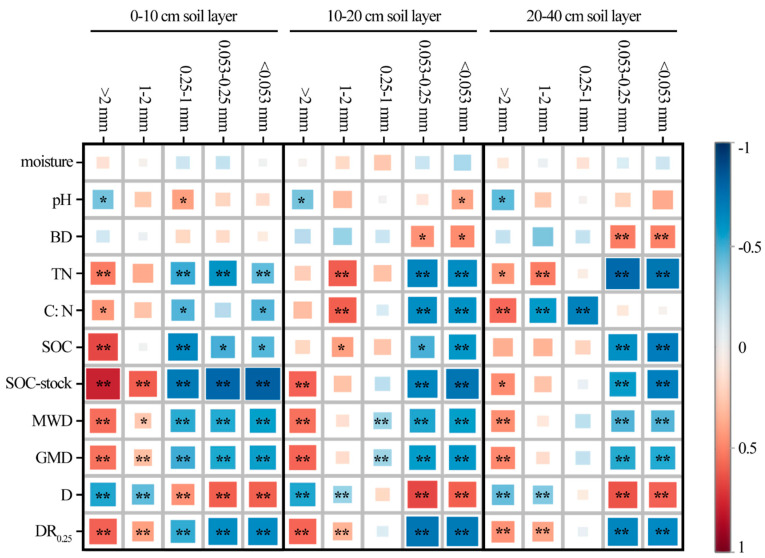
Pearson correlations between different soil aggregate sizes and soil physicochemical properties, including the soil moisture, pH, BD, TN, C:N ratio, SOC, SOC stock, and aggregate stability indexes, including MWD, GMD, DR_0.25_ and D, in different soil layers. **, when the confidence (double measurement) is 0.01, the correlation is obvious. *, when the confidence (double measurement) is 0.05, the correlation is obvious.

**Figure 5 microorganisms-11-02008-f005:**
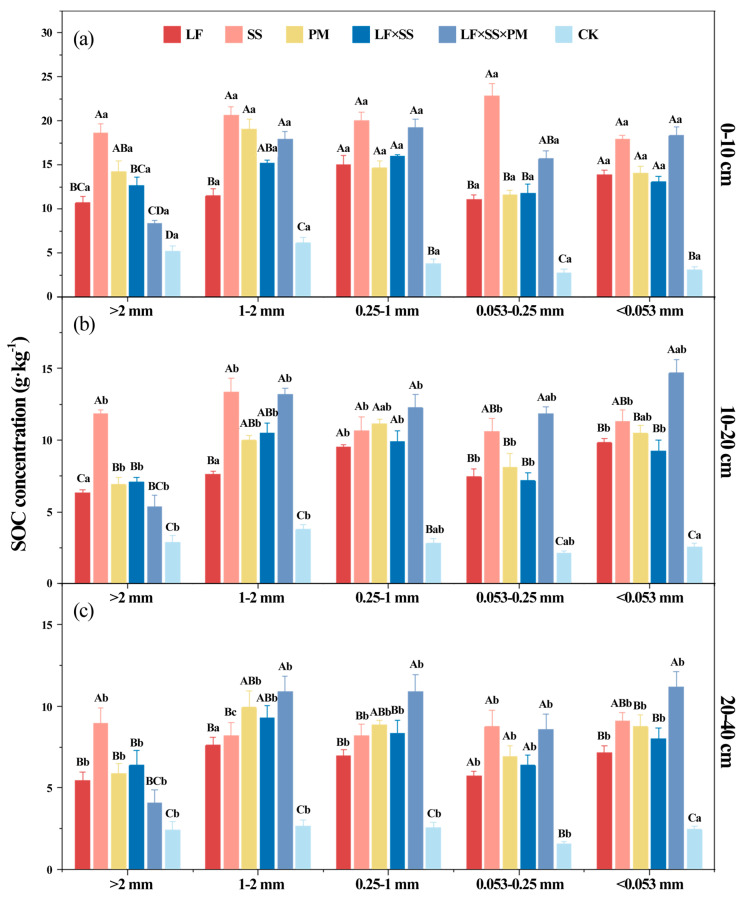
Differences in the soil aggregate organic carbon (C) at the 0–10 cm (**a**), 10–20 cm (**b**) and 20–40 cm (**c**) depths in five forest reforestation models and the CK in subtropical degraded red soil, showing the concentration of each particle size aggregate organic C. Different uppercase letters indicate differences between different tree species in the same soil layer, and different lowercase letters indicate differences between different soil layers in the same forest reforestation model (*p* < 0.05). LF, *Liquidambar formosana* pure forest; SS, *Schima superba* pure forest; PM, *Pinus massoniana* pure forest; LF × SS, mixed forest of *Liquidambar formosana* and *Schima superba*; LF × SS × PM, mixed forest of *Liquidambar formosana*, *Schima superba* and *Pinus massoniana*; CK, bare land.

**Figure 6 microorganisms-11-02008-f006:**
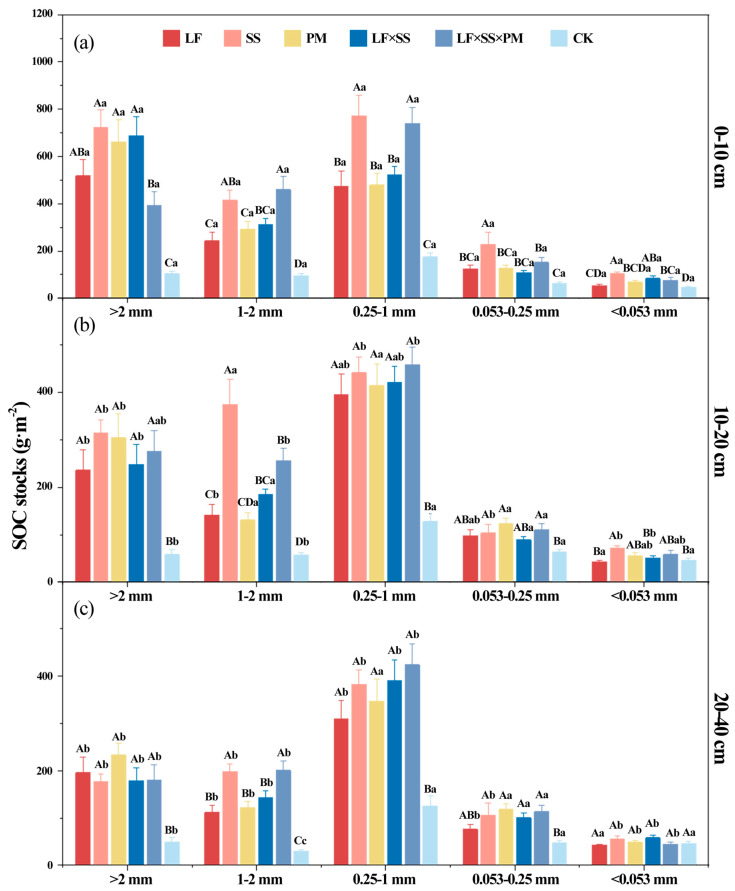
Differences in the soil aggregate organic carbon (C) stocks at the 0–10 cm (**a**), 10–20 cm (**b**) and 20–40 cm (**c**) depths in five forest reforestation models and the CK in subtropical degraded red soil, showing the concentration of organic C in each particle size aggregate. Different uppercase letters indicate differences between different tree species in the same soil layer, and different lowercase letters indicate differences between different soil layers in the same forest reforestation model (*p* < 0.05). LF, *Liquidambar formosana* pure forest; SS, *Schima superba* pure forest; PM, *Pinus massoniana* pure forest; LF × SS, mixed forest of *Liquidambar formosana* and *Schima superba*; LF × SS × PM, mixed forest of *Liquidambar formosana*, *Schima superba* and *Pinus massoniana*; CK, bare land.

**Figure 7 microorganisms-11-02008-f007:**
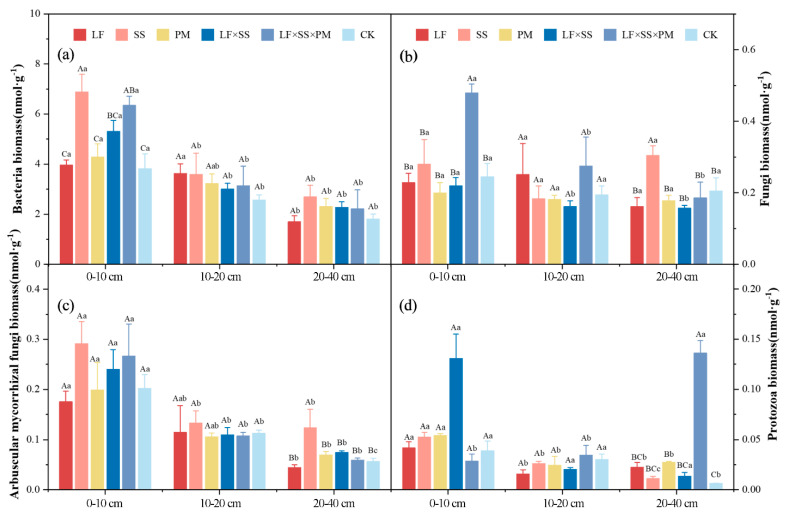
Soil microbial community composition biomass include bacterial biomass (**a**), fungi biomass (**b**), AMF biomass (**c**) and protozoa biomass (**d**) at the 0–10 cm, 10–20 cm and 20–40 cm soil layers in five forest reforestation models and the CK in subtropical degraded red soil. Different uppercase letters indicate differences between different tree species in the same soil layer, and different lowercase letters indicate differences between different soil layers in the same forest reforestation model (*p* < 0.05). LF, *Liquidambar formosana* pure forest; SS, *Schima superba* pure forest; PM, *Pinus massoniana* pure forest; LF × SS, mixed forest of *Liquidambar formosana* and *Schima superba*; LF × SS × PM, mixed forest of *Liquidambar formosana*, *Schima superba* and *Pinus massoniana*; CK, bare land.

**Figure 8 microorganisms-11-02008-f008:**
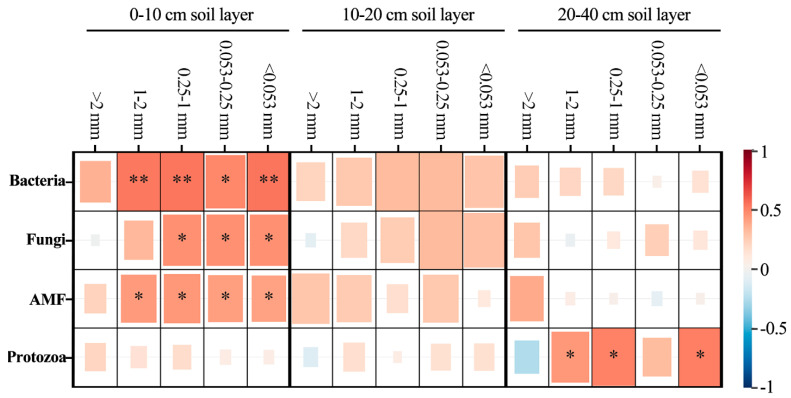
Pearson correlations between soil microbial biomass and the distribution of SOC in aggregates, including the soil bacteria, fungi, arbuscular mycorrhizal fungi (AMF), protozoa biomass. **, when the confidence (double measurement) is 0.01, the correlation is obvious. *, when the confidence (double measurement) is 0.05, the correlation is obvious.

**Table 1 microorganisms-11-02008-t001:** Basic situation of research sample plot.

Sample Numble	Reforestation Modle	Forest Proportion	Longitude/ Latitude	Elevation (m)	Forest Vegetation					
					Bush			Herb		
					Family	Genus	Species	Family	Genus	Species
1-1	LF		114°49′19″ N, 26°54′32″ E	114	8	10	10	7	8	8
1-2	LF		114°49′5″ N, 26°54′47″ E	114	3	3	3	2	2	2
1-3	LF		114°51′4″ N, 26°55′48″ E	113	9	11	11	10	11	11
1-4	LF		114°49′5″ N, 26°54′47″ E	118	6	7	7	8	10	10
2-1	SS		114°49′23″ N, 26°54′36″ E	109	5	5	5	6	6	6
2-2	SS		114°49′16″ N, 26°54′29″ E	102	4	4	4	5	5	5
2-3	SS		114°49′8″ N, 26°54′43″ E	87	0	0	0	0	0	0
2-4	SS		114°49′30″ N, 26°54′40″ E	87	0	0	0	0	0	0
3-1	PM		114°48′58″ N, 26°54′54″ E	124	4	6	6	7	7	7
3-2	PM		114°49′5″ N, 26°54′47″ E	111	5	6	7	7	7	7
3-3	PM		114°49′19″ N, 26°54′47″ E	102	7	7	7	5	5	5
3-4	PM		114°49′1″ N, 26°54′43″ E	109	0	0	0	0	0	0
4-1	LF × SS	LF:SS = 1:2	114°49′12″ N, 26°54′36″ E	108	8	8	9	6	6	6
4-2	LF × SS	LF:SS = 1:1	114°49′8″ N, 26°54′36″ E	113.7	7	7	8	5	5	5
4-3	LF × SS	LF:SS = 3:2	114°49′8″ N, 26°54′43″ E	109	8	10	11	8	9	9
4-4	LF × SS	LF:SS = 1:1	114°49′5″ N, 26°54′43″ E	102	5	5	5	6	6	6
5-1	LF × SS × PM	LF:SS:PM = 3:2:1	114°49′8″ N, 26°54′54″ E	97	8	8	8	4	6	6
5-2	LF × SS × PM	LF:SS:PM = 1:2:3	114°49′8″ N, 26°54′54″ E	106	5	5	5	2	2	2
5-3	LF × SS × PM	LF:SS:PM = 2:2:1	114°49′12″ N, 26°55′12″ E	80	7	7	7	3	5	5
5-4	LF × SS × PM	LF:SS:PM = 2:2:3	114°49′1″ N, 26°55′8″ E	76	7	7	7	5	6	6
6-1	CK		114°55′16″ N, 26°55′17″ E	82	0	0	0	0	0	0
6-2	CK		114°49′26″ N, 26°55′12″ E	81	0	0	0	0	0	0
6-3	CK		114°49′16″ N, 26°54′40″ E	109	0	0	0	0	0	0
6-4	CK		114°49′16″ N, 26°54′40″ E	108	0	0	0	0	0	0

Note: LF, *Liquidambar formosana* pure forest; SS, *Schima superba* pure forest; PM, *Pinus massoniana* pure forest; LF × SS, mixed forest of *Liquidambar formosana* and *Schima superba*; LF × SS × PM, mixed forest of *Liquidambar formosana*, *Schima superba* and *Pinus massoniana*; CK, bare land.

**Table 2 microorganisms-11-02008-t002:** Soil property comparison of different reforestation models.

Physicochemical Properties	Soil Layers	Moisture (%)	pH	BD (g·cm^−3^)	TN (g·kg^−1^)	C:N	SOC (g·kg^−1^)	SOC-Stock (kg·m^−2^)
LF	0–10 cm	0.12 ± 0.01 BCa	4.74 ± 0.02 Aa	1.19 ± 0.06 Ba	0.92 ± 0.01 Ba	22.23 ± 1.16 Ba	20.38 ± 1.05 ABa	2.56 ± 0.40 Aa
	10–20 cm	0.12 ± 0.01 Aab	4.63 ± 0.00 Ba	1.18 ± 0.04 Bab	0.67 ± 0.01 Bb	19.75 ± 0.90 Ab	13.16 ± 0.71 BCb	1.45 ± 0.24 Ab
	20–40 cm	0.13 ± 0.00 Ab	4.57 ± 0.01 Ca	1.16 ± 0.05 Bb	0.57 ± 0.02 Bc	13.92 ± 0.56 Bc	7.93 ± 0.32 Bc	0.82 ± 0.12 Ab
SS	0–10 cm	0.15 ± 0.01 ABa	4.73 ± 0.02 Aa	1.12 ± 0.03 ABa	1.05 ± 0.02 Aa	22.13 ± 0.34 Ba	23.10 ± 0.87 Aa	2.48 ± 0.34 Aa
	10–20 cm	0.13 ± 0.00 Ab	4.71 ± 0.01 Aa	1.14 ± 0.03 Ba	0.79 ± 0.01 Ab	15.24 ± 0.60 BCb	12.04 ± 0.55 ABb	1.34 ± 0.23 Ab
	20–40 cm	0.12 ± 0.00 Ab	4.71 ± 0.01 Aa	1.13 ± 0.03 Bb	0.59 ± 0.02 ABc	12.95 ± 0.79 BCc	7.69 ± 0.59 Bc	0.79 ± 0.12 Ab
PM	0–10 cm	0.12 ± 0.01 Ca	4.57 ± 0.04 Ba	1.16 ± 0.01 ABa	0.73 ± 0.02 Ca	29.54 ± 0.83 Aa	21.66 ± 0.48 Aa	2.49 ± 0.32 Aa
	10–20 cm	0.13 ± 0.01 Aa	4.57 ± 0.02 Ca	1.17 ± 0.03 Ba	0.67 ± 0.01 Bb	16.09 ± 0.66 Ab	10.72 ± 0.43 ABb	1.31 ± 0.09 Ab
	20–40 cm	0.13 ± 0.01 Aa	4.57 ± 0.01 Ca	1.15 ± 0.02 Bb	0.54 ± 0.02 Bc	17.12 ± 0.61 Ab	9.26 ± 0.60 Ac	1.05 ± 0.09 Ab
LF × SS	0–10 cm	0.15 ± 0.00 Aa	4.64 ± 0.01 Ba	1.16 ± 0.03 ABa	1.08 ± 0.01 Aa	20.27 ± 0.52 Ba	21.83 ± 0.68 ABa	2.40 ± 0.19 Aa
	10–20 cm	0.13 ± 0.01 Aa	4.63 ± 0.01 Bb	1.16 ± 0.02 Bb	0.75 ± 0.02 Ab	13.55 ± 0.17 Ca	10.17 ± 0.33 Cb	1.20 ± 0.11 Ab
	20–40 cm	0.13 ± 0.00 Aa	4.32 ± 0.01 Bc	1.16 ± 0.01 Bb	0.59 ± 0.02 ABc	12.08 ± 0.19 Cb	7.08 ± 0.10 Bc	0.86 ± 0.05 Ab
LF × SS × PM	0–10 cm	0.11 ± 0.01 Ca	4.73 ± 0.03 Aa	1.26 ± 0.06 Aa	0.73 ± 0.02 Ca	6.53 ± 0.33 Dc	4.72 ± 0.11 Ca	2.55 ± 0.14 Aa
	10–20 cm	0.11 ± 0.01 Aa	4.73 ± 0.01 Aa	1.23 ± 0.05 ABb	0.58 ± 0.01 Ca	8.05 ± 0.18 Da	4.69 ± 0.09 Da	1.42 ± 0.25 Ab
	20–40 cm	0.11 ± 0.01 Aa	4.72 ± 0.01 Aa	1.21 ± 0.05 ABb	0.63 ± 0.01 Cb	7.32 ± 0.12 Db	4.62 ± 0.09 Ca	0.82 ± 0.17 Ab
CK	0–10 cm	0.12 ± 0.01 BCa	4.74 ± 0.02 Aa	1.22 ± 0.02 ABa	0.63 ± 0.02 Db	9.23 ± 1.05 Ca	5.89 ± 0.80 Ca	0.61 ± 0.11 Ba
	10–20 cm	0.11 ± 0.01 Aa	4.73 ± 0.01 Aa	1.29 ± 0.03 Aa	0.45 ± 0.02 Da	6.40 ± 1.07 Dab	2.89 ± 0.52 Eb	0.37 ± 0.09 Ba
	20–40 cm	0.11 ± 0.01 Aa	4.73 ± 0.02 Aa	1.29 ± 0.02 Aa	0.47 ± 0.02 Db	3.98 ± 0.59 Eb	1.85 ± 0.21 Dc	0.30 ± 0.10 Ba

Note: Values shown are means ± standard errors, *n* = 16. Different uppercase letters indicate differences between different tree species in the same soil layer, and different lowercase letters indicate differences between different soil layers in the same forest reforestation model (*p* < 0.05), showing soil moisture, soil pH, soil bulk density (BD), total soil organic carbon (SOC), total soil nitrogen (TN) and the soil carbon and nitrogen ratio (C:N). LF, *Liquidambar formosana* pure forest; SS, *Schima superba* pure forest; PM, *Pinus massoniana* pure forest; LF × SS, mixed forest of *Liquidambar formosana* and *Schima superba*; LF × SS × PM, mixed forest of *Liquidambar formosana*, *Schima superba* and *Pinus massoniana*; CK, bare land.

**Table 3 microorganisms-11-02008-t003:** List of abbreviations.

Serial Number	Full Name of Indicator	Abbreviations	Unit
1	Soil bulk density	BD	g·cm^−3^
2	Total soil nitrogen	TN	g·kg^−1^
3	Total soil carbon and nitrogen ratio	C:N	
4	Total soil organic carbon	SOC	g·kg^−1^
5	Soil organic carbon stock	SOC-stock	kg·m^−2^
6	Large macroaggregates	>2 mm aggregates	%
7	Middle macroaggregates	1–2 mm aggregates	%
8	Small macroaggregates	0.25–1 mm aggregates	%
9	Microaggregates	0.053–0.25 mm aggregates	%
10	Clayey silt aggregates	<0.053 mm aggregates	%
11	Mean weight diameter	MWD	mm
12	Geometric mean diameter	GMD	mm
13	Content of >0.25 mm stable aggregates	DR_0.25_	%
14	Fractal dimension	D	
15	Large macroaggregate organic carbon	>2 mm aggregate-C	g·kg^−1^
16	Middle macroaggregate organic carbon	1–2 mm aggregate-C	g·kg^−1^
17	Small macroaggregate organic carbon	0.25–1 mm aggregate-C	g·kg^−1^
18	Microaggregate organic carbon	0.053–0.25 mm aggregate-C	g·kg^−1^
19	Clayey silt aggregate organic carbon	<0.053 mm aggregate-C	g·kg^−1^
20	Phospholipid fatty acid	PLFA	
21	Arbuscular mycorrhizal fungi	AMF	

## Data Availability

Data sharing not applicable. No new data were created or analyzed in this study. Data sharing is not applicable to this article.

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
