# Peer review of "Reforestation Increases the Aggregate Organic Carbon Concentration Induced by Soil Microorganisms in a Degraded Red Soil, Subtropical China"

_microorganisms, 2023, doi:10.3390/microorganisms11082008_

Round 1
Reviewer 1 Report
Dear Authors
aim and tasks good. Please specify the Aim distinctively in Introduction
major problems with referencing - error. source not found in text - dome digital problem?
Advice to add citation by Vincevica-Gaile "Towards Sustainable Stabilization..." 2021 as peatlands degraded ones with its soils provide major emission part from degraded lands per sei. There are some approaches for stabilization and circular economy
moderate check needed
Author Response
Response to Reviewer 1 Comments
Point 1: Please specify the Aim distinctively in Introduction
Response 1: In the last sentence of the introduction of the manuscript, we make clear the aim of this study. “The goal is to provide reference materials for soil quality evaluation and the soil erosion resistance of reforestation…”.
Point 2: Major problems with referencing - error. Source not found in text - dome digital problem?
Response 2: We have checked and modified the full text citation errors. Thank you for pointing out.
Point 3: Advice to add citation by Vincevica-Gaile "Towards Sustainable Stabilization..." 2021 as peatlands degraded ones with its soils provide major emission part from degraded lands per sei. There are some approaches for stabilization and circular economy.
Response 3: Thank you for your advice. We studied the literature content of Vincevica-Gaile and applied it in the introduction to support the feasibility of biological carbon sequestration in degraded soils. “Vincevica-Gaile proposed some methods of stable and circular economy by studying peatland degradation”
Vincevica-Gaile, Z.; Teppand, T.; Kriipsalu, M.; Krievans, M.; Jani, Y.; Klavins, M.; Setyobudi, R.H.; Grinfelde, I.; Rudovica, V.; Tamm, T. Towards Sustainable Soil Stabilization in Peatlands: Secondary Raw Materials as an Alternative. Sustainability 2021, 13.
Reviewer 2 Report
While the study is well compiled and present new insights there are several concerns as detailed
The abstract should be rewritten
- Grammatical error, quantitative information should be provided in the abstract
- The authors claimed that the soil quality improved after reforestation without providing numeric information to validate the claim
- Provide references to the first line in the introduction section
- List of abbreviations is required
- What is the criteria for selecting the 6 models are there previous study where the models have been applied
- A brief literature review would help readers understand the study novelty
- Check the citation error in the second paragraph of section 2
- Why was the location selected and would the developer model applicable to other locations ?
- In figure 1 all abbreviations and keys should be clear
- Reference citation error in lines 264 -266, this is common in the entire manuscript
- Please back the results of section 3 with literature and explain how they stand out
- The axis label in figure 2 should be enlarged
Needs improvement
Author Response
Response to Reviewer 2 Comments
Point 1: The abstract should be rewritten.
Response 1: Thank you for your advice. We have carefully reviewed the abstract and rewritten the parts that need to be improved to ensure that the expression is concise and accurate. We mainly revised the research methods and results in the abstract, focusing on the characteristics of the changes and the correlation of each factor.
Point 2: Grammatical error, quantitative information should be provided in the abstract.
Response 2: Through examination and further calculation, we supplement the quantitative data of the main conclusions in the abstract to support the research results., such as “the SOC stock increased by 131.28%-140.00%”,”...showed the largest proportion in macroaggregates (85.48%-89.37%)”.
Point 3: The authors claimed that the soil quality improved after reforestation without providing numeric information to validate the claim.
Response 3: Thank you for your reminder. Many research showed that the soil nutrients and structure of degraded red soil after reforestation were improved and enhanced, and the soil quality has been significantly improved. However, after examination and verification, we acknowledge that this expression in the manuscript is indeed somewhat general and vague. We have rewritten this part in the abstract to replace the important results obtained in this study. “After reforestation, the SOC stock increased by 131.28%-140.00%”.
Point 4: Provide references to the first line in the introduction section.
Response 4: The first line in the introduction has been added to the manuscript, with reference to the Liu 's literature (Liu et al., 2022), which explains that “approximately 50 billion metric tons of greenhouse gases are emitted every year”.
Liu, X.; Liu, G.; Xue, J.; Wang, X.; Li, Q. Hydrogen as a carrier of renewable energies toward carbon neutrality: State-of-the-art and challenging issues. Int. J. Miner., Metall. Mater. 2022, 29, 17.
Point 5: List of abbreviations is required.
Response 5: In the second section, we supplement the list of abbreviations appearing in the manuscript (Table 3). The table lists all the abbreviations and full names that appear in the manuscript, corresponding to the units used in this study.
Point 6: What is the criteria for selecting the 6 models are there previous study where the models have been applied.
Response 6: Reforestation and management will result in significant differences in SOC (Lichter et al., 2008; Grossiord et al., 2012). Studies have shown that higher tree species diversity in forests contributes to more SOC storage (Vesterdal et al., 2013); the SOC stock of broad-leaved forest is generally greater than that of coniferous forest, and the change of forest tree species composition has a long-term impact on soil structure and carbon storage (Su et,al., 2021). Therefore, how to use tree species for biological carbon sequestration needs to clarify the differences in soil carbon pools due to changes in broad-leaved or coniferous tree species.
Lichter J, Billings S A, Ziegler S E, et al. Soil carbon sequestration in a pine forest after 9 years of atmospheric CO2 enrichment. Global Change Biology 2008, 14(12): 2910-2922.
Grossiord C, Mareschal L, Epron D. Transpiration alters the contribution of autotrophic and heterotrophic components of soil CO2 efflux. New Phytologist 2012, 194(3): 647-653.
Vesterdal L, Clarke N, Sigurdsson B D, et al. Do tree species influence soil carbon stocks in temperate and boreal forests?. Forest Ecology and Management 2013, 309: 4-18.
Su F, Xu S, Sayer E J, et al. Distinct storage mechanisms of soil organic carbon in coniferous forest and evergreen broadleaf forest in tropical China. Journal of Environmental Management 2021, 295: 113142.
Point 7: A brief literature review would help readers understand the study novelty.
Response 7: Thank you for your suggestions, we have reviewed and combed the literature review of the manuscript, and we are doing our best to streamline and accurately express the novelty of this study.
Point 8: Check the citation error in the second paragraph of section 2.
Response 8: We have checked and corrected the citation errors in the second paragraph of section 2. Thank you for pointing out.
Point 9: Why was the location selected and would the developer model applicable to other locations ?
Response 9: Studies have shown that the soil erosion area in the red soil hilly area of southern China has reached 131,200 km2, accounting for about 15.10% of the national soil erosion area. Therefore, in the context of global climate change, strengthening soil conservation and ecological landscape restoration in red soil areas is one of the effective ways to solve the problem of ecological environment deterioration in the region. Since the 1980s, landscape reconstruction and soil erosion control have been vigorously carried out in the red soil area of southern China, and the comprehensive management of red soil in eroded areas has achieved remarkable results. By summarizing and analyzing the experience and practice, a relatively complete set of reforestation technology system suitable for the characteristics of low hilly red soil degradation area in Jiangxi Province was initially formed. After more than 30 years, the achievements of red soil erosion control have been remarkable, and some strategies have been put forward for the restoration of vegetation and ecological landscape in different types of soil erosion areas, hoping to help the restoration of more degraded soil in the future.
Point 10: In figure 1 all abbreviations and keys should be clear.
Response 10: The description of the specific location of the study area in the figure has been supplemented in the note of Figure 1 : “The study area is located in the degraded red soil in the low hilly area of Gousimaoling, Taihe County, Jiangxi Province, China”.
Point 11: Reference citation error in lines 264 -266, this is common in the entire manuscript.
Response 11: We have checked and modified all the citation errors in our manuscript, thank you for pointing out.
Point 12: Please back the results of section 3 with literature and explain how they stand out.
Response 12: Some studies have shown that SOC decomposition or preservation is related to the contact of decomposers and extracellular enzymes to SOC, and the role of soil microorganisms in the SOC operation process of forest soil ecosystems (Li et al., 2005) is crucial. It can be said that soil microorganisms are the core link in the carbon conversion process of forest ecosystems, and their composition characteristics (such as community composition, diversity, etc.) can affect the soil biofeedback regulation of forest carbon cycle (Spivak et al., 2019). As a good soil structure, aggregates are the key mechanism to protect SOC from microbial decomposition and keep SOC stability, and its effect is even stronger than the decomposability of SOC itself (Liu and Yu, 2011). Due to the low internal porosity of soil aggregates, air and water cannot enter directly, which hinders the decomposition of SOC by microorganisms, so that microorganisms can only decompose aggregates by secreting extracellular enzymes and consuming a lot of energy to decompose SOC (Cheng et al., 2019). However, at the same time, some soil microorganisms can promote the growth of individual plants, stabilize and increase the soil carbon pool by increasing the net primary productivity of forest ecosystems (Smith et al., 2003). Obviously, this is a contradictory part. Therefore, the third point of our study hypothesis is to demonstrate that microbial activity can promote the accumulation of SOC in degraded red soil during reforestation through the results.
Li Y, Xu M, Zou X, et al. Comparing soil organic carbon dynamics in plantation and secondary forest in wet tropics in Puerto Rico. Global Change Biology 2005, 11 (2): 239-248.
Spivak A C, Sanderman J, et al. Global-change controls on soil-carbon accumulation and loss in coastal vegetated ecosystems. Nature Geoscience 2019, 12 (9): 685-692.
Liu Z, Yu W.Research progress of organic carbon in soil aggregates. Chinese Journal of Ecological Agriculture 2011,19 (02): 447-455.
Chen Z, Zhou X, Geng S, et al. Interactive effect of nitrogen addition and throughfall reduction decreases soil aggregate stability through reducing biological binding agents. Forest Ecology and Management 2019, 445: 13-19.
Smith S, Smith F, Jakobsen I. Mycorrhizal fungi can dominate phosphate supply to plants irrespective of growth response. Plant Physiology 2003, 133: 16-20.
Point 13: The axis label in figure 2 should be enlarged.
Response 13: We examined all the diagrams in the manuscript and adjusted the inappropriate axis label, including Figure 2. Thank you for your reminder.
Round 2
Reviewer 1 Report
It is good to be published after changes